# Mesonephric-Like Adenocarcinoma of the Endometrium: Diagnostic Advances to Spot This Wolf in Sheep’s Clothing. A Review of the Literature

**DOI:** 10.3390/jcm10040698

**Published:** 2021-02-11

**Authors:** Ellen Deolet, Jo Van Dorpe, Koen Van de Vijver

**Affiliations:** 1Department of Pathology, Ghent University Hospital, 9000 Ghent, Belgium; ellen.deolet@uzgent.be (E.D.); jo.vandorpe@uzgent.be (J.V.D.); 2Cancer Research Institute Ghent (CRIG), Ghent University, 9000 Ghent, Belgium

**Keywords:** mesonephric-like adenocarcinoma, uterus, endometrium, histology, immunohistochemistry, molecular, *KRAS*

## Abstract

Mesonephric-like adenocarcinoma is a recently described rare neoplasm occurring in the uterine corpus and ovary. This under-recognized subtype of carcinoma can be very challenging to diagnose. In mesonephric adenocarcinoma a variety of growth patterns can be present within the same tumor, as a result of which they can be misinterpreted and diagnosed as low-grade endometrioid adenocarcinoma, clear cell carcinoma, or even serous carcinoma and carcinosarcoma. We report a case of mesonephric-like adenocarcinoma misdiagnosed as a low-grade endometrioid endometrial adenocarcinoma that had an early local recurrence and metastasized to the liver and the lungs. Histopathological, immunohistochemical and molecular analysis were performed and compared to published literature, providing a comprehensive overview of the current knowledge. Databases (Pubmed, Web of Science, Google Scholar) were searched with a combination of the following search terms: mesonephric-like, mesonephric, adenocarcinoma, carcinoma, uterine body, uterine corpus, endometrium. Mesonephric-like adenocarcinoma is a difficult-to-diagnose entity. Advanced diagnostics, including improved morphologic, immunohistochemical and molecular knowledge can help develop new therapeutic strategies against this specific subtype of endometrial cancer with an aggressive clinical behavior.

## 1. Introduction

Mesonephric-like adenocarcinomas (MLAs) represent recently described rare tumors occurring in the uterus and the ovaria. It is still a matter of controversy whether these tumors are of mesonephric origin or represent Müllerian neoplasms closely mimicking mesonephric adenocarcinomas. They show morphological, immunohistochemical and molecular similarities to mesonephric adenocarcinomas (MA) that originate from true mesonephric remnants. They have, however, overlapping features with Müllerian type carcinomas as an association with mesonephric remnants or hyperplasia is not always found and they arise in the endometrium. On a molecular basis, MA and MLA share *KRAS* mutations, but in MLA, concurrent *PIK3CA* mutations are described in nearly half of the cases [1], a mutation not found in MA and present among the genetic alterations in endometrioid adenocarcinoma [2]. There are a handful of cases where MLA is associated with other Müllerian neoplasms and clonality between the two is proven as they share identical *KRAS* or *NRAS* mutations [3,4]. Whole proteomic analysis, however, could not distinguish MA from MLA [5]. Most importantly, MLA is often misdiagnosed as other endometrial neoplasms, but has an aggressive clinical behavior and tend to metastasize early to the lungs [6,7,8,9,10]. 

## 2. Materials and Methods

We report a case of mesonephric-like adenocarcinoma misdiagnosed as a low-grade endometrioid endometrial adenocarcinoma that had an early local recurrence and metastasized to the liver and the lungs. Histopathological, immunohistochemical and molecular analyses were performed and compared to published literature, providing a comprehensive overview of the current knowledge. Databases (Pubmed, Web of Science, Google Scholar) were searched with a combination of the following search terms: mesonephric-like, mesonephric, adenocarcinoma, carcinoma, uterine body, uterine corpus, endometrium. After reading the title and the abstract, articles in English, where the full text was available, were included. Articles about all mesonephric-like adenocarcinomas (uterine body and ovarian) were included to give an overview of all the current knowledge. The search was performed by one of the authors (ED). 

## 3. Results

### 3.1. Case Presentation

A 76-year-old woman was referred to our hospital because of metastasized carcinoma. Her previous clinical history included a rectal adenocarcinoma 22 years ago (1998 TNM ‘98 pT3N1; treated with rectum resection and adjuvant chemotherapy with 5-FU and Elvorin). Three years ago she was diagnosed in another hospital with a low-grade (grade 1–2) endometrioid endometrial carcinoma with invasion to the inner half of the myometrium, TNM (UICC, 8th edition) pT1a, International Federation of Gynecology and Obstetrics (FIGO, 2009) stage IA. She underwent a total abdominal hysterectomy and bilateral salpingo-oophorectomy. She received adjuvant external radiation therapy of 46 Gy at the operation field, in the upper third of the vagina and elective lymph-node regions. Five months after primary surgery, she had a local recurrence at the vaginal vault, located in the irradiation field. This local recurrence was treated with simultaneous integrated boost radiation therapy (externally and brachytherapy) up to 62 Gy. Subsequently, 9 months after primary surgery, she developed liver metastasis, histologically consistent with the endometrial adenocarcinoma. The next metastases presented 4 months later, 1 year after primary surgery, both to the liver and the lung. This time she was not only treated with metastasectomy but also with carboplatin-taxol chemotherapy. Now, three years after primary surgery, she presents with a third metastasis, again to the lung.

All metastases showed similar morphology. They were composed of tubules and glandular structures with well-developed back-to-back glands and cribriform growth pattern. The cells showed moderate nuclear atypia with nuclear overlap, vesicular chromatin and an inconspicuous to the prominent eosinophilic nucleolus. The cytoplasm was slightly eosinophilic. There were numerous mitotic figures focally up to 6 mitoses/1HPF. Histology of the latest metastasis to the lung is depicted in Figure 1. Based on morphology, a diagnosis of metastatic well-differentiated endometrioid adenocarcinoma was considered. However, there was an inconsistency between the normally good prognosis of a low-grade endometrioid adenocarcinoma, part of the so-called WHO Type I tumors that are low-grade, estrogen-related, often clinically indolent, endometrioid carcinomas [11] and the early metastases in this case. Additional immunohistochemistry (see Figure 2) was performed, the tumor cells were positive for PAX8 and partly for CK7 and negative for SATB2 and CK20, confirming their gynecological and not colorectal origin. Estrogen receptor (ER) and progesterone receptor (PR) were negative, as were the previous metastases and the primary tumor. There was diffuse GATA3 positivity but no TTF1 expression. CD10 showed focal luminal positivity. The tumor cells exhibited a wild-type p53 expression. Mismatch repair protein (MMR) expression was preserved (MMR proficient). 

The primary resection was reviewed and showed an endometrial lesion of 2.3 cm with invasion into the inner myometrium. It had a glandular growth pattern with some papillary snouting and loss of polarity. The cells were columnar with enlarged oval nuclei, coarse chromatin and an eosinophilic nucleolus. Mitotic activity was not noticeably increased. There was no necrosis, no perineural invasion and no lymphovascular space invasion. No squamous or mucinous differentiation was seen. The right fallopian tube showed a micropapillary serous borderline tumor. The morphology of this cystic papillary serous lesion with mild cytonuclear atypia did not resemble a possible metastasis of the endometrial mesonephric-like adenocarcinoma (with different architecture, different chromatin pattern, different aspect of nucleoli). The left tube and both ovaries were normal.

With the morphology, supportive immunohistochemical profile and the history of early metastasis to a distant site, a diagnosis of a metastasis of a mesonephric-like adenocarcinoma was made. Additional molecular analysis showed a pathogenic *KRAS* c.38G > A variant (p.Gly13Asp) and two probable pathogenic variants of *PTEN* (c.388C > T and c.634 + 2T > G). 

### 3.2. Literature Search

Twenty-seven articles were found with a total of 154 case reports of MLAs (see Table 1, Table 2 and Table 3). Of these, there were 115 cases of the uterine body and 39 cases of the ovary. There were 12 cases included that originate in the myometrium, 72 in the endometrium and 31 not specified. A total of 16 tumors had associated findings. The clinical findings are listed in Table 1 and the microscopic, immunohistochemical and molecular findings in Table 2 and Table 3. For an overview of immunohistochemical findings, see Figure 3.

## 4. Discussion

### 4.1. Epidemiology

Mesonephric-like adenocarcinomas are rare neoplasms with a reported incidence of 1% of all endometrial carcinomas [7,8]. In the literature, 115 uterine and 39 ovarian cases have been reported. Of these, 16 MLA had associated findings of Müllerian origin: adenomyosis [22,23], endometriosis [4,22,31], atypical hyperplasia of the endometrium (or EIN) [10], serous cystadenoma [4], mixed serous and mucinous cystadenoma [4], serous borderline tumor [3,4,30], borderline endometrioid adenofibroma [4], low-grade endometrioid endometrial carcinoma [28], low-grade serous ovarian carcinoma [3,4]. 

All age groups were affected, ranging from 26 to 91 years with a mean of 59 years and a median of 61 years. 

### 4.2. Pathogenesis

The cell lineage of origin is still a matter of debate. With the morphology reminiscent of classic mesonephric carcinoma and overlapping immunohistochemical features, it could be a type of mesonephric carcinoma with divergent Müllerian features. Proteomic analysis of both MA and MLA was as good as identical [5]. On the other hand, uterine tumors tend to originate from the endometrium with secondary involvement of the myometrium and they are not associated with mesonephric remnants. Cases where the MLA is associated with other Müllerian lesions support the evidence of a Müllerian lesion that differentiated along the mesonephric lines. Yano et al., Dundr et al., McCluggage et al. and Chapel et al. could prove clonality between the Müllerian lesions (endometrioid endometrial carcinoma, serous borderline tumor and low-grade serous carcinoma of the ovary) since they share mutations in the *KRAS* and *NRAS* gene [3,4,28].

### 4.3. Morphology

MLA shows considerable overlap with conventional mesonephric carcinomas. They are characterized by a variety of growth patterns, between tumors and within the same tumor composed of small tubules, ductal/glandular growth, papillary, solid growth, sex cord-like, trabecular, retiform, sieve-like, glomeruloid and spindle cell areas are described. Luminal eosinophilic colloid-like secretions are characteristic but not always present. Ductular/glandular and tubular patterns are most frequently described (Table 2 and Table 3 [6]). The tumor cells may be flattened, cuboidal or columnar with usually scant eosinophilic cytoplasm. Focal cytoplasmic clearing is possible but rather rare [6,22]. There is mild to moderate cytological atypia. The nuclei can be oval to flattened, angulated with vesicular to optically clear chromatin, sometimes with nuclear groves or nuclear overlap. These nuclear features can be reminiscent of papillary thyroid carcinoma [6,22]. High-grade cytological atypia is normally not the predominant feature. Hobnail cells are a rarely reported feature [6,30].

There should be no squamous, ciliated or mucinous differentiation (metaplasia) and no associated mesonephric remnants.

### 4.4. Immunohistochemistry

MLAs are usually positive for Paired box protein-8 (PAX8), GATA binding protein 3 (GATA3), thyroid transcription factor 1 (TTF1), CD10 with luminal staining, and are negative for estrogen receptor (ER) and progesterone receptor (PR). However, focal positivity of ER is described by Kolin et al. (2/4 cases with weak, patchy or heterogeneous staining [8]), Pors et al. (2/5 cases weak to moderate in 10–55% [27]), Ando et al. (only a small number of cells (<1%) in the tubular pattern expressed ER [23]), Euscher et al. (6/23 cases with ER ranging from 10% up to 40% [6]), Kenny et al. [18], Patel et al. (weak focal staining in <5% [26]) and Yamamoto et al. (very focal [29]). Hence some positivity of ER does not preclude the diagnosis of MLA. PR was negative in all but two of these cases. So it can be concluded that PR is a more reliable negative marker for MLA. 

ER/PR negativity in endometrioid endometrial adenocarcinoma (EEC) is an independent risk factor for recurrence and death in FIGO grade I-II EEC [32]. However in these previous studies, no testing for GATA3 or TTF1 was performed, and so no definite conclusions can be drawn on how many of these ER/PR negative (low-grade) EEC constitute MLA. 

Calretinin, CD10 and ER used to be the markers to diagnose MLA before recognition of the role of TTF1 and GATA3. Howitt et al. compared GATA3 expression in mesonephric/Wolffian remnant with other tumors of the female genital tract. They found that GATA3 has a sensitivity of 98% and a specificity of 98% to differentiate mesonephric lesions to endocervical and endometrial carcinomas [20], which was confirmed by the whole proteome analysis by Gibbard et al. [5]. Later Pors et al. compared the sensitivity and specificity of GATA3, TTF1, CD10 and calretinin in the diagnosis of MLAs and reported GATA3 to be the best overall marker, but staining can be weak to moderate in intensity and positive in only a minority of cells (<10%) [27]. This finding was confirmed by Euscher et al. [6]. TTF1 and GATA3 regularly show an inverse staining pattern [6,26,27] and GATA3 is less expressed in more solid/spindled and sarcomatoid regions of the tumor [9,20,27]. 

CD10 shows in most of the cases at least focal expression with staining of the luminal/apical surface and has a reported sensitivity of 73% and specificity of 83% [10]. Calretinin positivity can support the diagnosis but is frequently negative. Moreover, CD10 is more difficult to interpret due to the positivity of surrounding endometrial stroma and smooth muscle and calretinin can give a background nonspecific granular cytoplasmic staining.

The expression pattern of p53 is wild type, p16 shows patchy staining, WT1 is negative. MLA is typically MMR proficient, with normal expression of MLH1, MSH2, MSH6 and PMS2. 

### 4.5. Molecular Findings

The majority of MLA harbor *KRAS* mutations, suggesting *KRAS* mutation is involved in MLA development. The *KRAS* mutation G12V and G12D are the most common, G12A and G12C are respectively 4 and 3 times reported. Concurrent *ARID1A* and *PIK3CA* mutations are relatively common and described in respectively nine and seven cases [1,6,9,30]. *PTEN* mutation, also frequent in EEC, was found as an additional mutation in three MLA cases [6,9]. In the case of Na et al., this was detected in metastatic tumor only, demonstrating that *PTEN* mutation is probably a relatively late event in the sequence of genetic alterations [9]. *KRAS* and *ARID1A* are common mutations in both MA and EEC, and so will not help in defining the mesonephric or Müllerian nature of MLA [2,33]. On the other hand, *PIK3CA* and *PTEN* mutations, which are common in EEC but have not been described in MA of the cervix are rather indicative of Müllerian origin with subsequent differentiation along mesonephric lines [33].

Copy number variation testing is increasingly being implemented. Copy number gain of 1q is most common [1,3,8,9,33] and some of these have also 1p loss. The gain of chromosome 10 is found in metastatic disease and may be an indicator of aggressive biological behavior [9,33].

Since these tumors have no aberrant p53 staining (no *TP53* mutation), have no loss of mismatch repair protein expression and so far have no POLE exonuclease domain hotspot mutation (*POLE*), they belong to the molecular group of no specific molecular profile (NSPM), and are probably responsible for the proportion of poor survivals in this group.

### 4.6. Prognosis

MLAs have aggressive biological behavior with more than half of the published cases presented with advanced stage (FIGO ≥ II) at diagnosis. They are associated with a considerable risk of recurrent disease with a tendency to metastasize to the lungs [6,7,8,9,10]. Not only high stage disease but also stage I disease frequently metastasizes [9,12,14,28,29,31]. This was confirmed by Pors et al. who calculated that the stage at diagnosis was not significant for progression-free survival. They reported a 5-year overall survival of 71 to 72% for mesonephric adenocarcinomas of the uterine body and ovary [10]. Six characteristics were significantly associated with the development of metastasis, including large tumor size (>4 cm), ill-defined tumor border, advanced FIGO stages (III to IV), presence of coagulative tumor cell necrosis, high mitotic activity (>10/10 high-power fields), and presence of lymphovascular invasion. These high mitotic activities and lymphovascular invasion were found to be independent factors [9]. Compared with other endometrial adenocarcinomas, MLAs have better overall survival than malignant mixed Müllerian tumors and serous carcinoma has equal overall survival to endometrioid grade 3 and has worse overall survival than endometrioid grade 1–2 carcinomas [10]. Endometrial carcinomas have a tendency for lymphovascular metastasis to pelvic lymph nodes followed by retroperitoneal lymph nodes. Distant metastases in endometrial carcinoma are rare with a reported incidence of 3.1% (all tumor types) [34]. Although the lungs are the most common site (1.5%), only 1.1% of EEC do present with lung metastasis [34].

### 4.7. Treatment

All cases were treated with a total hysterectomy and bilateral salpingo-oophorectomy. Pelvic lymph node dissection was often added, potentially also with para-aortic lymph node dissection. Adjuvant chemotherapy, mainly carboplatin + paclitaxel, was given in high stage disease but also in one case of FIGO stage IA and two cases of FIGO stage IB [9,23]. Radiation therapy was given solo in early cases and in addition to chemotherapy in higher stage cases. Two reported cases were treated with hormone therapy [6,28]. One case was diagnosed as a low-grade EEC, the concurrent MLA component was only retrospectively recognized, and was treated with progesterone therapy; 6 years later only the MLA recurred [28]. The other case was also diagnosed as EEC, grade 1–2, the type of hormone therapy was not specified. This tumor recurred with distant metastasis to the liver after 17 months [6].

The optimal regimen and the efficacity of (neo) adjuvant radiation and or chemotherapy remains largely unknown. So far, no tumor-specific treatment options have been elucidated for MLA.

### 4.8. Differential Diagnosis

The diagnosis can be challenging due to the rarity and with the diverse histologic pattern, the tumor is also frequently under-recognized and misdiagnosed. There is some degree of morphologic overlap with EEC, clear cell carcinoma, serous carcinoma, as well as carcinosarcoma. Tubules with eosinophilic secretions are a diagnostic clue for MLA. Of course one must exclude cervical mesonephric adenocarcinoma with the involvement of the uterine corpus. This can be done macroscopically by determining where the tumor is predominantly located and needs to be correlated to imaging. Microscopically, mesonephric carcinomas of the cervix are frequently associated with mesonephric remnants/hyperplasia, while this should not be seen in MLA. Most cases of MLA of the uterine body originate in the endometrium and cases described as MA of the uterine corpus originate more in the lateral walls in the myometrium, where mesonephric remnants are to be expected. In this review tumors that originate in the myometrium but where no mesonephric remnants near the tumor were found are also included for completeness of mesonephric lesions in the uterine corpus, but these could also be true MA instead of MLA (see italics in Table 1 and Table 2). Both MA and MLA have negative ER and PR staining and express GATA3. Calretinin and CD10 may also be positive in both MLA and MA. MLAs have in comparison to MA more frequently TTF1 positive staining [27]. For further differentiation, additional molecular testing can be performed. MLA and MA share KRAS mutations, but when PTEN and *PIK3CA* mutations are found the diagnosis of MLA is made above a MA.

The main differential diagnosis is EEC with the tubular and glandular growth pattern. MLA has tubules and glands as the most common feature but is characterized by common heterogeneity of architectural patterns. Cytological features of nuclei with vesicular chromatin and nuclear grooves seen in MLA are not characteristic of endometrial carcinoma, while EECs are usually composed of cells that are columnar with pseudostratified nuclei. When squamous, ciliated or mucinous differentiation is seen, MLA can be excluded. Endometrial hyperplasia or endometrioid intraepithelial neoplasia are precursor lesions for EEC, however there are reports where these are found with MLA (including the case presented in this manuscript) [10], but lack of these favors MLA. Immunohistochemically, MLA is characterized by GATA-3 and/or TTF1 expression, which is rare in EEC [20,27]. EECs are normally positive for estrogen and progesterone receptors, which are nearly always absent in MLA, with negative PR as the most reliable marker. So when diffuse and strong ER and PR are found, a diagnosis of MLA is very unlikely. GATA3 expression can be seen in a minority of endometrial carcinomas: 6% reported by Pors et al., including endometrioid adenocarcinomas, but these cases were always TTF1 negative [10,20]. Terzik et al. reported GATA3 positivity in endometrial premalignant and malignant proliferations with an incidence of 8% (5 of 64 cases): one with atypical hyperplasia, one high-grade endometrioid adenocarcinoma, two serous carcinomas and one carcinosarcoma. Additionally, GATA3 expression in EECs is not diffuse but focal to patchy with weak to strong staining [35]. For TTF1 Pors et al. reported only 1.0% (6 of 585 cases) of endometrial neoplasms with TTF1 expression, including three endometrioid carcinomas, one serous carcinoma, one clear cell carcinoma and one carcinosarcoma [27]. Other earlier studies reported TTF1 expression ranging from 2% up to 19% in EEC and 9–23% in serous carcinomas and 7% clear cell carcinoma [36,37]. These EECs with TTF1 expression are reported to have a worse prognosis [36]. It could be possible that some of these may actually represent MLA since the cases of Ervine et al. [36] were all ER-negative and there was no additional immunohistochemical staining for GATA3, PR, CD10 or calretinin reported.

When the papillary architectural pattern in association with high-grade nuclear atypia is observed, serous endometrial carcinoma should be considered. Serous carcinoma is characterized by p53 mutation and p16 block-staining, which is never seen in MLA. ER and PR are not helpful in differentiating because both MLA and serous carcinomas share negative hormone receptor expression.

Clear cell carcinomas can also have a combination of architectural patterns, with variable cytological atypia and a low mitotic index. Hobnail cells and cytoplasmic clearing are seldom seen in MLA but can raise the possibility of clear cell carcinoma. Immunohistochemically, clear cell carcinomas are typically positive for HNF-1B, and often for napsin A and/or Alpha methyacyl CoA racemase (AMACR). Napsin A and HNF-1b are mostly negative in MLA, but can be positive. Clear cell carcinomas also show negative ER/PR staining, but can have abnormal p53 and can be mismatch repair deficient, in contrast to MLA that always have wild type p53 staining and the reported cases are mismatch repair proficient. GATA3 and TTF1 are usually negative in clear cell carcinomas.

With solid areas and spindled cell and sarcomatoid features, MLA can be confused with carcinosarcoma. Lack of heterologous differentiation, as well as wild type p53 staining, suggests the tumor is less likely a carcinosarcoma. Moreover, p53 wild-type carcinosarcomas often demonstrate microsatellite instability (MSI) (rather representing undifferentiated or dedifferentiated carcinomas), and do not appear to harbor *KRAS* mutations [38]. 

In a metastatic setting, like in our case, with metastasis to the lungs, the positive TTF1 staining and negative hormone receptors can be confusing with primary lung adenocarcinoma. In patients with a history of gynecological malignancy one should always perform PAX8 staining. The morphology of pseudoendometrioid glands and small glands with eosinophilic secretions give a clue for MLA. Additional GATA3 staining, CD10 and calretinin staining can further help to support the diagnosis of MLA.

## 5. Conclusions

Features that should make the pathologist think about the possibility of MLA are the presence of a combination of architectural patterns, with most frequently ductular/glandular and tubular growth pattern, in a tumor without squamous of mucinous differentiation. A diagnostic clue is the presence of intraluminal dense eosinophilic secretions. When such morphology is observed, additional immunohistochemical staining can be performed. We suggest using PAX8, GATA3, TTF1, ER and PR as first-line markers, as proposed by Pors et al. 2018 [27]. When positive, CD10 and calretinin might be helpful additional markers. The tumor should be MMR proficient. Molecular analysis with the finding of *KRAS* and possibly *PIK3CA*, *ARID1A* or *PTEN* mutation can support the diagnosis.

Further investigation is needed for endometrial tumors that have a loss of hormone receptors and positive staining for GATA3 and/or TTF1, with p53 wild type pattern and MMR proficient. Are all these tumors, with histology that fits MLA as well as other subtypes of endometrial tumors, by definition MLA?

Mesonephric-like adenocarcinomas are considered high-grade carcinomas, even though they have a misleadingly low-grade morphology. The tumors have a high risk of recurrence and a high tendency for lung metastasis. Further research on the pathogenesis should help better understand this specific subset of endometrial cancer. As of today, tumor-specific treatment options are limited and the best therapeutic strategy is yet to be determined. 

## Figures and Tables

**Figure 1 jcm-10-00698-f001:**
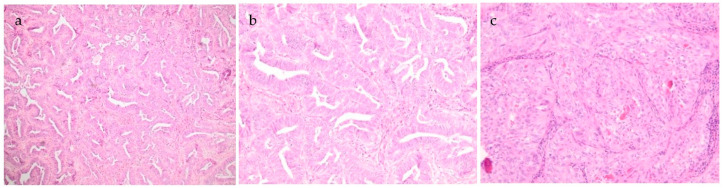
(**a**–**c**). Metastasis to the lung with glandular and ductular to focal solid growth pattern. Several eosinophilic intraluminal secretions are present (**c**). No high-grade atypia and low mitotic figures. (magnification: 40×, 100×, 100×-HE staining).

**Figure 2 jcm-10-00698-f002:**
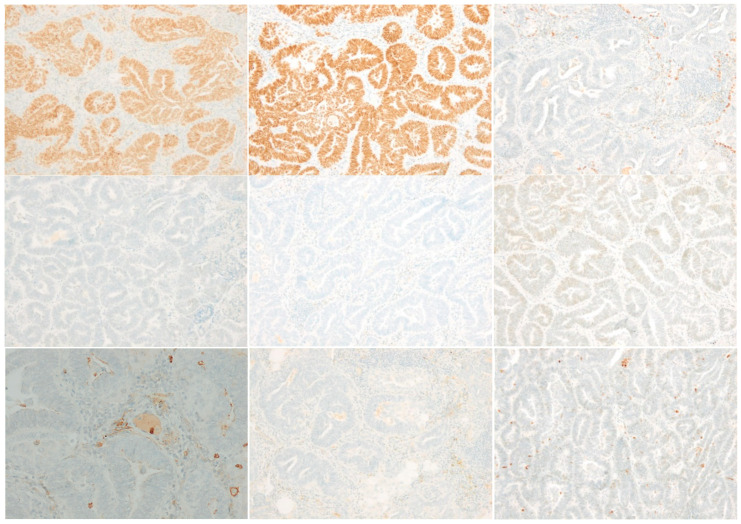
Diffuse positive staining for PAX8 and GATA3. TTF1, SATB2, estrogen receptor (ER) and progesterone receptor (PR) are negative (PR shows minor non-specific cytoplasmic background staining). CD10 focal luminal positivity. Calretinin negative; p53 wild type staining. (magnification: 100×; immunohistochemistry).

**Figure 3 jcm-10-00698-f003:**
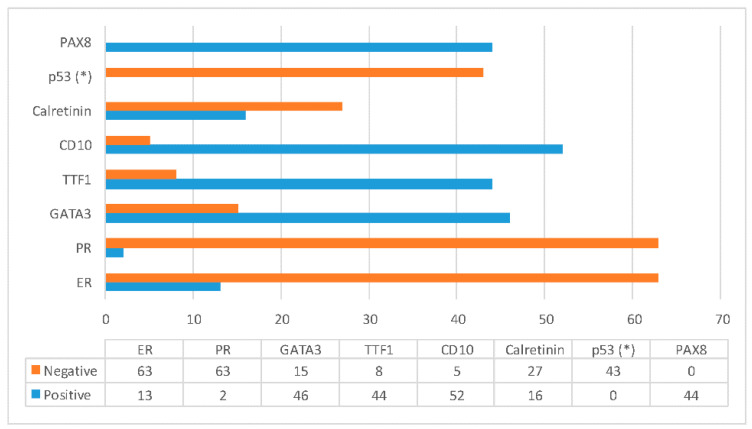
Summary of immunohistochemical findings. Only cases where information was available of individual markers are included. Estrogen receptor (ER) and progesterone receptor (PR) are nearly always negative, with PR as a more reliable negative marker. Mesonephric-like adenocarcinomas (MLAs) are characterized by staining with GATA binding protein 3 (GATA3) and/or thyroid transcription factor 1 (TTF1) with sometimes a reversed staining pattern. CD10 was predominantly positive in the tested cases but mostly focal. Calretinin is more often negative. All cases showed Paired box protein-8 (PAX8) positivity and p53 wild type expression. (*) = negative for TP53 mutation.

**Table 1 jcm-10-00698-t001:** Clinical findings.

	Case	Age	Location	Associated Findings	FIGO Stage 2009	Treatment	Chemotherapy Y/N	Radiation Y/N	Recurrence Y/N (Location R/Treatment)	Follow-Up Time (mo)
*1*	Yamamoto et al., 1995 [12]	*58*	Myometrium	*Cervical Gartner duct cyst*	*IA **	*TH + BSO*	*N*	*N*	*Y (paraaortic and mediastinal lymph node metastases R/cisplatinum + cyclophosphamide)*	*8 DED*
*2*	Ordi et al., 2001 [13]	*33*	Myometrium	*None*	*IA **	*TH + BSO + PLND*	*N*	*Y*	*N*	*8 NED*
*3*	Montagut et al., 2003 [14]	*33*	Myometrium	*None*	*IB **	*Myomectomy and subsequent TH + BSO + PLND + PALND*	*N*	*Y*	*Y (peritoneal carcinomatosis and lung metastasis R/carboplatin + paclitaxel)*	*22 AWD*
4	Bague et al., 2004 [15]	37	Endometrium	None Diagnosed as MMMT	IA *	TH + BSO	N	N	N	45 NED
*5*	Marquette et al., 2006 [16]	*81*	Myometrium to endometrium	*None*	*IA **	*TH + BSO + PLND + PALND*	*N*	*N*	*N*	*9 NED*
*6*	Wani et al., 2008 [17]	*73*	Myometrium to endometrium	*None*	*IVB*	*TH + BSO*	*Y carboplatin + paclitaxel*	*N*	*Y (lung metastases and abdominal disease R/carboplatin + paclitaxel)*	*28 AWD*
*7*	Kenny et al., 2012 [18]	*NA*	Myometrium	*Benign mesonephric remnants in the cervix*	*IIIA*	*TH + BSO*	*NA*	*NA*	*NA*	*NA*
*8*	Wu et al., 2014 [19]	*55*	Myometrium	*None*	*IB*	*TH + BSO + PLND*	*N*	*N*	*N*	*7 NED*
*9*	Wu et al., 2014 [19]	*62*	Myometrium	*None*	*IB*	*TH + BSO + PLND*	*N*	*N*	*N*	*1 NED*
10	Howitt et al., 2015 [20]	NA	NA	NA	NA	NA	NA	NA	NA	NA
11	Kim et al., 2016 [21]	66	Myometrium to endometrium	Adenomyosis	IB	TH + BSO + PLND	N	N	N	2 NED
12–23	McFarland et al., 2016 [22]	42–72	7 uterine corpus 5 ovarian	Endometriosis in 3 of 5 (ovarian cases) Adenomyosis (1/7)	IA(3 cases), IB(1 case), IIIC(1 case)	NA	NA	NA	N (IA, IB cases) Y (IIIC case)	18 (IA, IB cases) 56 (IIIC case)
*24*	*Ando* et al., *2017* [23]	*61*	Myometrium	*Adenomyosis*	*IB*	*TH + BSO*	*Y carboplatin + paclitaxel*	*N*	*N*	*9 NED*
25	Kim et al., 2018 [24]	37	Endometrium to myometrium	None	IB	TH + BSO	NA	NA	NA	NA
26	Zhang et al., 2019 [25]	63	Endometrium to myometrium	None	IB	TH + BSO+ PLND + PALND	N	Y	N	31 NED
27	Zhang et al., 2019 [25]	57	Myometrium	None	IIIB	TH + BSO + PLND	Y	N	NA	NA
28	Chapel et al., 2018 [3]	80	Ovary	Serous borderline tumor and low-grade serous carcinoma	not reported	TH + BSO + omentectomy + tumor debulking	Y Neoadjuvant carboplatin + paclitaxel	N	N	3 NED
29	Patel et al., 2018 [26]	71	Endometrium to myometrium	None	IB	TH + BSO	NA	NA	NA	NA
30	Pors et al., 2018 [27]	65	Endometrium	None	IVB	NA	NA	NA	NA	NA
31	Pors et al., 2018 [27]	31	Endometrium	None	IIIA	NA	NA	NA	NA	NA
32	Pors et al., 2018 [27]	75	Endometrium	None	IB	NA	NA	NA	NA	NA
33	Pors et al., 2018 [27]	91	Endometrium	None	IIIA	NA	NA	NA	NA	NA
34	Pors et al., 2018 [27]	67	Ovary	None	IC	NA	NA	NA	NA	NA
35	Na et al., 2019 [9]	58	Endometrium to myometrium	None	IIIB	TH + BSO + PLND + PALND	Y carboplatin + paclitaxel	Y	Y (lung R/carboplatin + paclitaxel	56 AWD
36	Na et al., 2019 [9]	55	Endometrium to myometrium	None	IVB	TH + BSO	Y carboplatin + paclitaxel	Y	Y (carboplatin + paclitaxel)	21 AWD
37	Na et al., 2019 [9]	54	Endometrium to myometrium	None	IIIB	TH + BSO	Y carboplatin + paclitaxel	Y	Y (lung R/doxorubicin + cisplatin)	20 AWD
38	Na et al., 2019 [9]	60	Endometrium to myometrium	None	IA	TH + BSO + PLND + PALND	Y	N	Y (lung R/carboplatin + paclitaxel)	14 AWD
38	Na et al., 2019 [9]	53	Endometrium to myometrium	None	IA	TH + BSO + PLND + PALND	N	N	N	12 NED
40	Na et al., 2019 [9]	57	Myometrium	None	IIIC	TH + BSO + PLND + PALND	Y carboplatin + paclitaxel	N	Y (lung R/carboplatin + paclitaxel)	13 AWD
41	Na et al., 2019 [9]	70	Endometrium to myometrium	None	IB	TH + BSO + PLND + PALND	N	Y	N	10 NED
42	Na et al., 2019 [9]	61	Endometrium to myometrium	None	IB	TH + BSO + PLND + PALND	Y carboplatin + paclitaxel	N	N	7 NED
43	Na et al., 2019 [9]	65	Endometrium to myometrium	None	IB	TH + BSO + PLND + PALND	N	N	N	6 NED
44	Na et al., 2019 [9]	59	Endometrium to myometrium	None	IA	TH + BSO + PLND + PALND	N	N	N	11 NED
45	Na et al., 2019 [9]	52	Endometrium to myometrium	None	IIIC	TH + BSO + PLND + PALND	Y carboplatin + paclitaxel	Y	N	3 NED
46	Yano et al., 2019 [28]	32	Endometrium to myometrium	Low grade endometrioid carcinoma	IA	TH + BSO+ omentectomy	medroxyprogesterone acetate	N	Y 6 year later	NA
47	Kolin et al., 2019 [8]	64	Endometrium to myometrium	None	IB	TH + PLND + omentectomy	NA	NA	Y local (3y) lung metastases (12y R/carboplatin/taxol)	150 AWD
48	Kolin et al., 2019 [8]	57	Endometrium to myometrium	None	IA	NA	NA	NA	N	18 NED
49	Kolin et al., 2019 [8]	58	Endometrium to myometrium	None	IVB	NA	NA	NA	Y (local)	30 AWD
50	Kolin et al., 2019 [8]	62	Endometrium to myometrium	None	IIIC	NA	NA	NA	Y (lung)	100 DOD
51	Yamamoto et al., 2019 [29]	70	Endometrium to myometrium	None	IA	TH	NA	NA	Y (lung (5y))	NA
52	McCluggage et al., 2020 [4]	61	Ovary	Serous borderline tumor (low-grade serous carcinoma in extraovarian tissues)	IIIA1	NA	Y carboplatin + paclitaxel	N	NA	NA
53	McCluggage et al., 2020 [4]	66	Ovary	Borderline endometrioid adenofibroma	NA	NA	NA	NA	NA	NA
54	McCluggage et al., 2020 [4]	77	Ovary	Endometriosis, mixed serous and mucinous cystadenoma	NA	NA	NA	NA	NA	NA
55	McCluggage et al., 2020 [4]	50	Ovary	None	NA	NA	NA	NA	NA	NA
56	McCluggage et al., 2020 [4]	73	Ovary	Serous cystadenoma	NA	NA	NA	NA	NA	NA
57	Dundr et al., 2020 [30]	61	Ovary	Serous borderline tumor	IV	TH + BSA + resection of liver metastases and the diaphragm, total omentectomy, appendectomy and a resection of an umbilical metastasis	Y Neoadjuvant: carboplatin + paclitaxel Postoperative: carboplatin + paclitaxel + bevacizumab	N	N	12 NED
58	Seay et al., 2020 [31]	67	Ovary	Endometriosis	IA	TH + BSO + PLND + omentectomy	N	N	Y (abdominal 18 mo R/carboplatin+ paclitaxel +bevacizumab)	18 AWD
59–102	Pors et al., 2020 [10] 44 cases	28–91	Endometrium	2 cases with complex atypical hyperplasia (EIN)	I (18/43) II-IV (25/43 cases)	NA	NA	NA	Y (24/41) lungs (14/22), pelvis (5/24), liver (3/24), vagina (3/24), brain (2/24), spleen (2/24), abdomen (2/24), omentum (1/24), peritoneum (1/24), abdominal wall (1/24), vertebrae (1/24)	1 to 130 months (mean 44 mo). 5-y PFS 27.5% 5-year OS/DSS 72%
103–127	Pors et al., 2020 [10] 25 cases	36–81	Ovary	None	I (11/18) II-IV (7/18)	NA	NA	NA	Y (10/24 cases) lungs (2/5), omentum (2/5), liver (1/5), iliopsoas (1/5), pubic bone (1/5), perihepatic region (1/5), mesentery (1/5), and peritoneum (1/5).	1 to 1346 months (mean 101 mo). 5-year PFS 68% 5-year OS/DSS 71%
128–131	Horn et al., 2020 [7] 4 cases	54–74	Endometrium to myometrium	None	IB (2/4) IIIA (1/4) IIIC (1/4)	NA	NA	NA	NA	NA
132–154	Euscher et al., 2020 [6] 23 cases	26–75	Uterus	None	I (11/23) II (1/23) III (7/23) IV (4/23)	NA	I: 5 RT; 1 CT; 2 RCT; 1 hormone therapy; 1 N; 1 NA II: 1 RCT III: 1 RT; 1 CT; 5 RCT IV: 3 CT; 1 NA		Y (lung 9/23; liver 2/23; abdomen 1/23, pelvis 1/23, vagina 1/23, unknown location (1/23)) N (4/23) NA (2/23) Never free of tumor (2/23) (7 cases of stage I disease with distant recurrence)	NED (4/23, 3–74 mo) AWD (10/23, 9–121 mo) DOD (6/23, 20–83 mo)
155	Our case	73	Endometrium to myometrium	None	IA	TH + BSO	N	Y 45Gy	Y 5 mo local R/RT; 9 mo liver R/metastasectomy; 12 mo liver and lung R/metastasectomy + CT carbo/taxol 36 mo Lung	36 NED

TH: total hysterectomy; BSO: bilateral salpingo-oophorectomy; PLND: pelvic lymph node dissection; PALND: para-aortic lymph node dissection; NA: not available/not done; AWD: alive with disease; NED: no evidence of disease; DOD: died of disease; RT: radiation therapy; CT: chemotherapy; RCT: radiation + chemotherapy; mo: month(s); Y: yes; N: no. * = FIGO stage adjusted to 2009 classification.

**Table 2 jcm-10-00698-t002:** Morphologic, immunohistochemical and molecular findings.

	Case	Tubular	Papillary	Glandular	Solid	Spindle Cell	Retform	Eosinophilic Secretions	Atypia	Other
*1*	*Yamamoto* et al., *1995* [12]	*x*	*x*		*x*	*x*			*Severe.*	*Frequent mitotic figures.*
*2*	*Ordi* et al., *2001* [13]	*x*	*x*	*x*	*x*	*x*	*x*	*x*	*Moderate.*	*Mitotic index 3/HPF.*
*3*	*Montagut* et al., *2003* [14]	*x*		*x*		*x*	*x*	*x*	*NA*	*NA*
4	Bague et al., 2004 [15]		x	x	x	x		x		
*5*	*Marquette* et al., *2006* [16]	*x*	*x*	*x*			*x*	*x*		
*6*	*Wani* et al., *2008* [17]	*x*	*x*	*x*			*x*	*x*	*Mild to moderate.*	*Glomeruloid*
*7*	*Kenny* et al., *2012* [18]	*x*	*x*	*x*	*x*	*x*		*x*	*Focally severe.*	*Sex cord like* *Focal mitotic activity*
*8–9*	*Wu* et al., *2014* [19] *2 cases*	*x*		*x*	*x*	*x*	*x*	*x*		*Sex cord like*
10	Howitt et al., 2015 [20]									
11	Kim et al., 2016 [21]	x	x	x			x	x	Mild.	Glomeruloid Mitotic index >10/10HPFs
12–23	McFarland et al., 2016 [22] 12 cases + Mirkovic et al., 2018 [1] for molecular analysis (7 cases of McFarland et al.)	x	x	x	x			x	Moderate.	Slit-like Mitotic activity conspicuous.
*24*	*Ando* et al., *2017* [23]	*x*	*x*	*x*	*x*			*x*	*Mild.* *More atypical component in the periphery.*	*Occasional mitotic figures.*
25	Kim et al., 2018 [24]	x		x				x	Severe, yet monomorphic nuclei	
26	Zhang et al., 2019 [25]	x	x		x	x		x	Bland cuboidal cells	
27	Zhang et al., 2019 [25]	x	x		x		x	x		
28	Chapel et al., 2018 [3]	x			x			x	Mild to moderate.	Psammomatous calcifications focally Mitotic index 1–2/10 HPFs.
29	Patel et al., 2018 [26]	x	x	x				x	Mild	Prominent stromal hyalinization Mitotic activity was relatively low level without atypical mitotic figures
30–34	Pors et al., 2018 [27] 5 cases	Classic morphological features of mesonephric carcinoma but occurring outside the cervix and without mesonephric remnants								
35–45	Na et al., 2019 [9] 11 cases	x	x	x		x	x		Mild to moderate (7/11) Severe (4/11).	Glomeruloid Sex cord like Mitotic activity 4–23/10 HPFs (mean 13/10HPFs, median 15/10HPFs).
46	Yano et al., 2019 [28]	x	x	x	x					
47	Kolin et al., 2019 [8]									
48	Kolin et al., 2019 [8]		x	x		x				
49	Kolin et al., 2019 [8]									
50	Kolin et al., 2019 [8]									
51	Yamamoto et al., 2019 [29]	x	x		x		x	x		
52(53–56NA)	McCluggage et al., 2020 [4]	x		x	x			x	Moderate	Mitotic figures were easily identified Focal areas of necrosis
57	Dundr et al., 2020 [30]	x			x			x		Mitotic index of 4 mitoses/10 HPFs
58	Seay et al., 2020 [31]	x	x	x	x				Moderate	Corded growth Increased mitotic activity
59–127	Pors et al., 2020 [10] 44 + 25 cases	Classic morphological features of mesonephric carcinoma but occurring outside the cervix and without mesonephric remnants.								
128–131	Horn et al., 2020 [7] 4 cases	x	x	x	x				Low grade	Mean mitotic index of 26.5/10HPFs (range 21–33).
132–154	Euscher et al., 2020 [6] 23 cases	x	x	x	x	x	x	x	Mild to moderate Focally marked	Cords / trabeculae Sieve like Mitotic index 3–28/10 HPFs (median: 10)
155	Our case	x		x					Moderate	Mitotic index 2–6/1HPF.

Italics: located in the myometrium. NA: not available; HPF: High power field; x: present; +: positive; −: negative.

**Table 3 jcm-10-00698-t003:** Morphologic, immunohistochemical and molecular findings.

	Case	Immunohistochemistry (IHC)					Molecular
		ER/PR	GATA3	TTF-1	CD10	Other/Remarks	
*1*	*Yamamoto* et al., *1995* [12]	*NA*	*NA*	*NA*	*NA*	*/*	*NA*
*2*	*Ordi* et al., *2001* [13]	*−/−*	*NA*	*NA*	*+*	*P53 wild type* *Inhibin –*	*NA*
*3*	*Montagut* et al., *2003* [14]	*−/−*	*NA*	*NA*	*+*	*P53 wild type*	*NA*
4	Bague et al., 2004 [15]	NA	NA	NA	Not specified	Case diagnosed as MMMT	NA
*5*	*Marquette* et al., *2006* [16]	*−/−*	*NA*	*NA*	−	*/*	*NA*
*6*	*Wani* et al., *2008* [17]	*−/−*	*NA*	*NA*	*+ luminal*	*Calretinin +* *P53 wild type* *HNF1b –*	*NA*
*7*	*Kenny* et al., *2012* [18]	*Not specified (2/8+)*	*NA*	*Not specified (3/5 +)*	*Not specified (6/8 +)*	*Report of 8 cases, 7 cervical, 1 uterine body; IHC is not separately reported.*	*NA*
*8–9*	*Wu* et al., *2014* [19] *2 cases*	*−/−*	*NA*	*NA*	*+*	*Calretinin +*	*NA*
10	Howitt et al., 2015 [20]	NA	+	NA	NA	/	NA
11	Kim et al., 2016 [21]	−/−	NA	+	+ luminal	Calretinin – Inhibin – P53 wild type	NA
12–23	McFarland et al., 2016 [22] 12 cases + Mirkovic et al., 2018 [1] for molecular analysis (7 cases of McFarland et al.)	−/− (12/12)	+ (3/12) − (8/12)	+ (11/12) − (1/12)	+ (7/9) − (2/9)	P53 wild type Calretinin + (3/6) HNF1b − (8/10) NapsinA − (9/11)	*KRAS* mutations 7/7 G12D (4/7) G12V (3/7) *PIK3CA* activating mutations (3/7). There were no alterations in *PTEN*, *ARID1A*, or *TP53* in any of the tumors. CNV: 1q gain (5/7), accompanied by 1p loss in 2 cases. Chromosome 10 gain (4/7), which was accompanied by gain of chromosome 12 in 3 cases.
*24*	*Ando* et al., *2017* [23]	*+ (<1%)/*−	*+*	*+*	*+ luminal*	*Calretinin + focal* *WT1 –* *Inhibin –* *NapsinA + (<1%)* *P53 wild type*	*NA*
25	Kim et al., 2018 [24]	−/−	+	NA	+	/	NA
26	Zhang et al., 2019 [25]	−/−	+	NA	+ luminal	Calretinin + focal WT1 + focal	NA
27	Zhang et al., 2019 [25]	−/−	+	−	NA	WT1 – NapsinA – P53 wild type P16 patchy	NA
28	Chapel et al., 2018 [3]	−/−	+	+	+ luminal	Calretinin – P53 wild type P16 patchy WT1 – Inhibin + focal P63 + focal Thyroglobulin – CK20 – PTH – Chromogranin – Synaptophysin –	*NRAS* Q61R additional mutations in the tumor suppressor genes *BCOR* or *AMER1* CNV: 1q gain, 18p gain, 1p loss, 18q loss, 22 loss.
29	Patel et al., 2018 [26]	+ (weak focal<5%)/−	+	+	NA	Beta-catenin nuclear Inverse staining of GATA3/TTF1	*KRAS* G12A
30–34	Pors et al., 2018 [27] 5 cases	+/ND (2/5) −/ND (3/5)	+ (5/5)	+ (5/5)	+ luminal (4/5)	Inverse staining of GATA3/TTF1 Calretinin –	NA
35–45	Na et al., 2019 [9] 11 cases	−/− (11/11)	+ (11/11)	NA	+ luminal (11/11)	P53 wild type Preserved PTEN Calretinin + (3/11)	11 cases + from 1 case a metastasis. *KRAS* mutation (10/12) G12V (6/10) G12C (2/10) G12D (2/10) *ARID1A* mutation (9/12) T294P (6/9) Q288P (2/9) Q287Pfs (1/9) 1q gain (11/12) 2 of the cases with 1q gain also had loss of 1p. 9p gain (7/12), 20q gain (7/12), 12q gain (6/12), 6q gain (4/12), 10q gain (4/12), 3q loss (3/12), 5p gain (3/12), 7q gain (3/12), 19p gain (3/12), and gain of chromosome 2 (3/12). Gain of 10q was detected exclusively in 3 cases with metastasis. Additional *PTEN* mutation (D268E) in metastatic tumor only.
46	Yano et al., 2019 [28]	−/−	+ focal	+ focal	+ focal	P53 wild type CA125 strong P16 focal Calretinin – HNF1b – NapsinA – AR – WT1 –	*KRAS* G12A
47	Kolin et al., 2019 [8]	+ (patchy, weak)/−	+ focal	+	+ luminal	Synaptophysin – Chromogranin – P63 –	*KRAS* G12A 1p loss, 1q gain, 10p gain, 10q loss, 21 q loss
48	Kolin et al., 2019 [8]	+/+ heterogenous	−	+	+ luminal	P53 wild type	*KRAS* G12V 1p loss, 1q gain, 4p loss, 4q loss, 11p loss, 11q loss, 21q loss
49	Kolin et al., 2019 [8]	−/−	−	+ patchy	+ luminal	P16 patchy NapsinA focal +	*KRAS* G12D *PIK3R1* E451del 1q gain, 11p loss, 11q loss, 13q loss, 17p loss, 22q loss
50	Kolin et al., 2019 [8]	−/−	NA	+	NA	P53 wild type NapsinA – WT1 – Chromogranin – Synaptophysin – Thyroglobulin –	*KRAS* G12V
51	Yamamoto et al., 2019 [29]	+ very focal/−	+ focal	+ diffuse	+ focal	Calretinin focal + Thyroglobulin -	NA
52(53–56NA)	McCluggage et al., 2020 [4]	−/−	+ focal	+ focal	+ luminal	P53 wild type WT1 – Thyroglobulin –	*KRAS* G12D
57	Dundr et al., 2020 [30]	−/−	+ (30%)	+ (70%)	+ focal	P53 wild type Calretinin – WT1 – HNF1b – Inhibin –	*KRAS* (c.34G > T, p.(G12C)) and *PIK3CA* (c.1633G > A, p.(E545K)) Likely pathogenic somatic *MYCN* mutation (c.131C > T, p.(P44L) Hereditary *CHEK2* mutation.
58	Seay et al., 2020 [31]	−/−	+ focal	+ focal	+ focal	P16 patchy WT1 – Calretinin – P53 wild type MMRp Preserved PTEN, ARID1A	Variants of unknown significance in the *ATM* gene (c.4303A > C (p.Lys145Gln) and *PALB2* gene (c.693A > T (p.Lys231Asn)
59–127	Pors et al., 2020 [10] 44 + 25 cases	Positive for at least one of GATA3, TTF1, CD10 (luminal), calretinin, AND negative/focal positivity for ER, or molecular confirmation (*KRAS* mutations)					
128–131	Horn et al., 2020 [7] 4 cases	−/− (4/4)	− (2/2)	+ (4/4)	+ luminal (2/2)	P16 patchy P53 wild type MMRp	*KRAS* G12V (3/4) *KRAS* G12D (1/4)
132–154	Euscher et al., 2020 [6] 23 cases	−/− (11/23) +/- (3/23) +/+ (1/23) +/ND (2/23) −/ND (4/23) ND/ND (2/23) (ER + 10–40% focal to patchy)	GATA3+/TTF1+ (10/23) GATA3+/TTF1-(4/23) GATA3-/TTF1+ (0/23) GATA3-/TTF1-(1/23) GATA3 NA/TTF1+ (1/23) GATA3+/TTF1 NA (1/23) NA (6/23)		+ luminal (10/23) NA (13/23)	Calretinin − (10/15) + (5/15)	13/17 *KRAS* G12D (7/13) G12V (5/13) G12A (1/13) Five cases with *KRAS* mutation also had additional mutations including *PIK3CA* (3/5); *PTEN* (2/5) and *CTNNB1* (1/5).
155	Our case	−/−	+	−	−	P53 wild type PAX8 + SATB2 – CK7 focal + CK20 focal + P16 patchy Vimentin + MMRp	*KRAS* G13N Two probable pathogenic variants of *PTEN* (c.388C > T and c.634 + 2T > G).

Italics: located in the myometrium. ND: not done; NA: not available; MMRp: mismatch repair proficient; +: positive; −: negative.

## Data Availability

No new data were created or analyzed in this study. Data sharing is not applicable to this article.

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
