# Peer review of "Mesonephric-Like Adenocarcinoma of the Endometrium: Diagnostic Advances to Spot This Wolf in Sheep’s Clothing. A Review of the Literature"

_jcm, 2021, doi:10.3390/jcm10040698_

Round 1
Reviewer 1 Report
There are minor changes suggested in the attached document
- 'And' is redundant in the author list
- The sentence in the abstract 'classic mesonephric adenocarcinoma with a variety of growth patterns that can be present within the same tumor. As a result...' would be better phrased as
'in mesonephric adenocarcinoma a variety of growth patterns that can be present within the same tumor as a result of which they can be misinterpreted...' - Line 39 add the word present as indicated Mutation not found in MA and present among
- Line 75 - were the metastases present in lung or liver or both?
- Line 102 - spelling mistakes in borderline and tube
- Line 102 - what was the IHC of the tubal borderline tumour? Was consideration given to likely metastasis from MA?
- Line 195 - considering the tendency of MA to metastasise to the lung, a greater description of the differences would be worthwhile
- Line 268 - Discussion of IHC differences between MA and MLA would be good
- Line 281- Greater emphasis on the consistent lack of staining with ER is advised
- Line 285/6 - Is there any data on GATA3 staining of EEC?

Author Response
Response to Reviewer 1 Comments
Thank you very much for your favourable response on our manuscript entitled “Mesonephric-like Adenocarcinoma of the Endometrium: Diagnostic Advances to Spot This Wolf in Sheep’s Clothing. A Review of the Literature.” We highly appreciate that you are willing to review the manuscript after revision.
We have revised the manuscript according to the suggestions of the reviewers. We also performed an additional English spell check. We are looking forward to hearing your decision on our revised manuscript. Please do not hesitate to contact us in case of any questions.
Point 1
'And' is redundant in the author list
Author response 1
We thank the reviewer for pointing this out and deleted the word ‘and’.
Point 2
The sentence in the abstract 'classic mesonephric adenocarcinoma with a variety of growth patterns that can be present within the same tumor. As a result...' would be better phrased as
'in mesonephric adenocarcinoma a variety of growth patterns that can be present within the same tumor as a result of which they can be misinterpreted...'
Author response 2
We agree that the sentence is not well phrased. We changed the sentence as suggested by the reviewer to:
“This underrecognized subtype of carcinoma can be very challenging to diagnose. In mesonephric adenocarcinoma a variety of growth patterns can be present within the same tumor, as a result of which they can be misinterpreted and diagnosed as low-grade endometrioid adenocarcinoma, clear cell carcinoma, or even serous carcinoma and carcinosarcoma.”
Point 3
Line 39 add the word present as indicated Mutation not found in MA and present among
Author response 3
We thank the reviewer for this suggestion and added the word ‘present’.
Point 4
Line 75 - were the metastases present in lung or liver or both?
Author response 4
There were metastases present in the liver and in the lungs. The latest metastatic presentation was only to the lungs. All had the same morphology. We changed the manuscript accordingly to make this more clear:
“All metastases showed similar morphology. They were composed of tubules…”
Point 5
Line 102 - spelling mistakes in borderline and tube
Author response 5
We thank the reviewer for pointing this out. We corrected the spelling.
Point 6
Line 102 - what was the IHC of the tubal borderline tumour? Was consideration given to likely metastasis from MA?
Author response 6
We thank the reviewer for this important remark, as it could indeed have changed the original FIGO staging of the primary tumor, with possible therapeutic consequences. When receiving the lung metastasis in 2020, we reviewed the primary diagnosis of her rectal adenocarcinoma of 1998 and of the endometrial adenocarcinoma of 2017 and even without IHC we were convinced it was a borderline tumor, indeed being aware of the possibility of a metastasis from the MLA. We added the following to the manuscript:
“The morphology of this cystic papillary serous lesion with mild cytonuclear atypia did not resemble a possible metastasis of the endometrial mesonephric-like adenocarcinoma (with different architecture, different chromatin pattern, different aspect of nucleoli).”
Point 7
Line 195 - considering the tendency of MA to metastasise to the lung, a greater description of the differences would be worthwhile
Author response 7
We thank the reviewer for this excellent remark. We added a paragraph in the differential diagnosis for metastatic setting (lines 343-348 in resubmitted article):
“In metastatic setting, like in our case, with metastasis to the lungs, the positive TTF1 staining and negative hormone receptors can be confusing with primary lung adenocarcinoma. In patients with a history of a gynecological malignancy one should always perform a PAX8 staining. The morphology of pseudoendometrioid glands and small glands with eosinophilic secretions give a clue for MLA. Additional GATA3 staining, CD10 and calretinin staining can further help to support the diagnosis of MLA.”
We also added a paragraph on how (in)frequent other endometrial carcinomas metastasize to the lungs (lines 255-259 in resubmitted article):
“Endometrial carcinomas have a tendency for lymphovascular metastasis to pelvic lymph nodes followed by retroperitoneal lymph nodes. Distant metastases in endometrial carcinoma is rare with a reported incidence of 3,1% (all tumor types) [38]. Although the lungs are the most common site (1,5%), only 1,1% of EEC do present with lung metastasis [38].”
Point 8
Line 268 - Discussion of IHC differences between MA and MLA would be good
Author response 8
We agree that this would be good to describe and added the following in the section of differential diagnosis at the end of the first paragraph (lines 285-295 in resubmitted article):
“Most cases of MLA of the uterine body originate in the endometrium and cases described as MA of the uterine corpus originate more in the lateral walls in the myometrium, where mesonephric remnants are to be expected. In this review tumors that originate in the myometrium but where no mesonephric remnants near the tumor were found are also included for completeness of mesonephric lesions in the uterine corpus, but these could also be true MA instead of MLA (see italics in tables 1 and 2). Both MA and MLA have negative ER and PR staining and express GATA3. Calretinin and CD10 may also be positive in both MLA and MA. MLA have in comparison to MA more frequently TTF1 positive staining [17]. For further differentiation additional molecular testing can be performed. MLA and MA share KRAS mutations, but when PTEN and PIK3CA mutations are found the diagnosis of MLA is made above a MA.”
Point 9
Line 281- Greater emphasis on the consistent lack of staining with ER is advised
Author response 9
Indeed when there is diffuse and strong ER staining a MLA can be excluded (lines 306-308 in resubmitted article). We changed the manuscript accordingly:
“EEC are normally positive for estrogen and progesterone receptors, which are nearly always absent in MLA, with negative PR as most reliable marker. So when diffuse and strong ER and PR is found, a diagnosis of MLA is very unlikely.”
Point 10
Line 285/6 - Is there any data on GATA3 staining of EEC?
Author response 10
Yes there is, this was described in the part of immunohistochemistry, but might indeed better be placed in the part of differential diagnosis. We added the data (lines 309-323 in resubmitted article):
“GATA3 expression can be seen in a minority of endometrial carcinomas, 6% reported by Pors et al., including endometrioid adenocarcinomas, but these cases were always TTF1 negative [10, 22]. Terzik et al. reported GATA3 positivity in endometrial premalignant and malignant proliferations with an incidence of 8% (5 of 64 cases): one with atypical hyperplasia, one high grade endometrioid adenocarcinoma, two serous carcinomas and one carcinosarcoma. Also GATA3 expression in EEC is not diffuse but focal to patchy with weak to strong staining [23]. For TTF1, Pors et al. reported only 1.0% (6 of 585 cases) of endometrial neoplasms with TTF1 expression, including three endometrioid carcinomas, one serous carcinoma, one clear cell carcinoma and one carcinosarcoma [17]. Other earlier studies reported TTF1 expression ranging from 2% up to 19% in endometrioid adenocarcinomas and 9-23% in serous carcinomas and 7% clear cell carcinoma [36-37]. These endometrioid carcinomas with TTF1 expression are reported to have worse prognosis [36]. It could be possible that some of these may in actually represent MLA since the cases of Ervine at al. [36] were all ER negative and there was no additional immunohistochemical staining for GATA3, PR, CD10 or calretinin reported.”
Reviewer 2 Report
The manuscript of “Mesonephric-like Adenocarcinoma of the Endometrium: Diagnostic Advances to Spot This Wolf in Sheep’s Clothing. Review of the Literature” was reviewed. MLA is recently described as a rare tumor in uterine corpus and ovary, which is not well known in pathogenesis, diagnosis, treatment and prognosis. This literature was well summarized and described in detail in MLA based on handful previous reports. This manuscript has large potential of helpfulness for clinical and pathological physicians encountered this tumor. The manuscript is well written. There are some minor comments as follows,
- In case presentation, line 5. FIGO should not be abbreviated at the first represent in the manuscript.
- Where was the local recurrence? Out of the first irradiation fields? Please show the precise location of the recurrence.
- In figure 2, please show what are indicate a, b, and c.
- Was genomic analysis examined using whole tumor tissue? Or microscopic captured tissue?
- Summarize of Immunohistochemical examination in all reported cases is interesting. These summations would be helpful to avoid misdiagnosis to low grade endometrial carcinoma and another confusing uterine corpus carcinoma and ovarian carcinoma. Please create tables indicating dynamics of each molecules in immunohistochemistry.
- In molecular analysis, it is interesting that PIK3CA an ARID1A mutations were some MLA cases, which generally similar with endometrioid carcinoma, not associated in MA. Please show possible explanation of these findings and mechanism. And in the speculation that PTEN mutation would be added lately in metastatic disease, please show the reference and rationale.
Author Response
Response to Reviewer 2 Comments
The manuscript of “Mesonephric-like Adenocarcinoma of the Endometrium: Diagnostic Advances to Spot This Wolf in Sheep’s Clothing. Review of the Literature” was reviewed. MLA is recently described as a rare tumor in uterine corpus and ovary, which is not well known in pathogenesis, diagnosis, treatment and prognosis. This literature was well summarized and described in detail in MLA based on handful previous reports. This manuscript has large potential of helpfulness for clinical and pathological physicians encountered this tumor. The manuscript is well written. There are some minor comments as follows.
Author response
Thank you very much for your kind words and your favourable response on our manuscript entitled “Mesonephric-like Adenocarcinoma of the Endometrium: Diagnostic Advances to Spot This Wolf in Sheep’s Clothing. A Review of the Literature.” We highly appreciate that you are willing to review the manuscript after revision.
We have revised the manuscript according to the suggestions of the reviewers. We also performed an additional English spell check. We are looking forward to hearing your decision on our revised manuscript. Please do not hesitate to contact us in case of any questions.
Point 1
In case presentation, line 5. FIGO should not be abbreviated at the first represent in the manuscript.
Author response 1
We thank the reviewer for pointing this out. We added “International Federation of Gynecology and Obstetrics stage (FIGO, 2009)”.
Point 2
Where was the local recurrence? Out of the first irradiation fields? Please show the precise location of the recurrence.
Author response 2
This local recurrent was in the irradiation field and located at the vaginal vault. We completed the location of the irradiation field. The patient received a boost at the location of the recurrence up to 62 Gy (so in this location 16 Gy more). We added the following to the manuscript (lines 67-70):
“Five months after primary surgery she had a local recurrence at the vaginal vault, located in the irradiation field. This local recurrence was treated with simultaneous integrated boost radiation therapy (externally and brachytherapy) up to 62 Gy.”
Point 3
In figure 2, please show what are indicate a, b, and c.
Author response 3
The legend of figure 1 was unfortunately not visible due to layout artefacts, and hidden under the pictures of figure 2. Enters were added to make it reappear. The original figure legend of figure 1 is:
“Figure 1 (a,b,c). Metastasis to the lung with glandular and ductular to focal solid growth pattern. Several eosinophilic intraluminal secretions are present (c). No high grade atypia and low mitotic figures. (magnification: 40x, 100x, 100x - HE staining).”
Point 4
Was genomic analysis examined using whole tumor tissue? Or microscopic captured tissue?
Author response 4
Molecular analysis of the tumor was performed using the routine technique of macrodissection. Based on the microscopic H&E slide, the surrounding healthy tissue is removed to enrich the sample for tumor cells. This technique and it usefulness in routine practice has been described in multiple papers, e.g. by de Bruin et al. BMC Genomics 2005 Oct 14;6:142. doi: 10.1186/1471-2164-6-142 (hyperlink: https://pubmed.ncbi.nlm.nih.gov/16225673/). We decided not to add this to the manuscript, but are always willing to do so if the reviewer feels it is necessary.
Point 5
Summarize of Immunohistochemical examination in all reported cases is interesting. These summations would be helpful to avoid misdiagnosis to low grade endometrial carcinoma and another confusing uterine corpus carcinoma and ovarian carcinoma. Please create tables indicating dynamics of each molecules in immunohistochemistry.
Author response 5
We thank the reviewer for this excellent suggestion and we provided an overview of useful immunohistochemical markers depicted in a single table, so we added Table 3. In the revised manuscript we added the following to line 126:
“For an overview of immunohistochemical findings see Table 3.”
Point 6 part 1
In molecular analysis, it is interesting that PIK3CA an ARID1A mutations were some MLA cases, which generally similar with endometrioid carcinoma, not associated in MA. Please show possible explanation of these findings and mechanism.
Author response 6 part 1
We thank the reviewer for giving this remark. We also noticed a small error in our text because ARID1A mutations have been described in MA, just like KRAS mutations (see reference 24 Mirkovic et al. 2015). We changed the manuscript (lines 223-230) to explain how these findings can help elucidating the mechanisms of pathogenesis of mesonephric-like adenocarcinoma, in comparison to mesonephric carcinoma and endometrial endometrioid carcinoma.
“KRAS and ARID1A are common mutations in both MA and EEC, and so will not help in defining the mesonephric or Müllerian nature of MLA [2, 24]. On the other hand PIK3CA and PTEN mutations, which are common in EEC but have not been described in MA of the cervix, are rather indicative of a Müllerian origin with subsequent differentiation along mesonephric lines [24].”
Point 6 part 2
And in the speculation that PTEN mutation would be added lately in metastatic disease, please show the reference and rationale.
Author response 6 part 2
We described this in paragraph 4.6 Molecular findings. “PTEN mutation, also frequent in EEC, was found as an additional mutation in three MLS cases [6, 9]. In the case of Na et al. this was detected in metastatic tumor only, demonstrating that PTEN mutation is probably a relatively late event in the sequence of genetic alterations [9].”
So only in the article written by Na et al. [reference 9] it was described that the PTEN mutation was found in the metastasis only and not in the primary tumor. These authors suggested the probable hypothesis that this PTEN mutation might be a relatively late event.